# Lineage Conversion in Pediatric B-Cell Precursor Acute Leukemia under Blinatumomab Therapy

**DOI:** 10.3390/ijms23074019

**Published:** 2022-04-05

**Authors:** Alexandra Semchenkova, Ekaterina Mikhailova, Alexander Komkov, Marina Gaskova, Ruslan Abasov, Evgenii Matveev, Marat Kazanov, Ilgar Mamedov, Anna Shmitko, Vera Belova, Anna Miroshnichenkova, Olga Illarionova, Yulia Olshanskaya, Grigory Tsaur, Tatiana Verzhbitskaya, Natalia Ponomareva, Gleb Bronin, Konstantin Kondratchik, Larisa Fechina, Yulia Diakonova, Liudmila Vavilova, Natalia Myakova, Galina Novichkova, Alexey Maschan, Michael Maschan, Elena Zerkalenkova, Alexander Popov

**Affiliations:** 1Dmitry Rogachev National Medical Research Center of Pediatric Hematology, Oncology and Immunology, 117998 Moscow, Russia; semalex94@mail.ru (A.S.); katmikhailova1805@gmail.com (E.M.); alexandrkomkov@yandex.ru (A.K.); st.mira@mail.ru (M.G.); ruslan.abasov.2013@gmail.com (R.A.); hen9.matveeff@yandex.ru (E.M.); mkazanov@gmail.com (M.K.); imamedov78@gmail.com (I.M.); anna.miroshnichenkova@gmail.com (A.M.); olga.illarionoff@gmail.com (O.I.); yuliaolshanskaya@gmail.com (Y.O.); yulia.diakonova@gmail.com (Y.D.); mila.vavilova@mail.ru (L.V.); nmiakova@mail.ru (N.M.); gnovichkova@yandex.ru (G.N.); amaschan@mail.ru (A.M.); mmaschan@yandex.ru (M.M.); eazerkalenkova@gmail.com (E.Z.); 2Department of Genomics of Adaptive Immunity, Shemyakin-Ovchinnikov Institute of Bioorganic Chemistry, 117998 Moscow, Russia; 3Institute for Information Transmission Problems (the Kharkevich Institute, RAS), 127051 Moscow, Russia; 4Skolkovo Institute of Science and Technology, 121205 Moscow, Russia; 5Center for Precision Genome Editing and Genetic Technologies for Biomedicine, Pirogov Russian National Research Medical University, 119334 Moscow, Russia; annashmi97@gmail.com (A.S.); verusik.belova@gmail.com (V.B.); 6Regional Clinical Children Hospital, 620149 Ekaterinburg, Russia; tsaur@mail.ru (G.T.); uralverba@gmail.com (T.V.); lfechina@mail.ru (L.F.); 7Research Institute of Medical Cell Technologies, 620026 Ekaterinburg, Russia; 8Russian Children Clinical Hospital, 119571 Moscow, Russia; ignata.ponomareva@gmail.com; 9Morozov City Children Clinical Hospital, 119049 Moscow, Russia; gleb-bronin@ya.ru (G.B.); kondratchik@mail.ru (K.K.)

**Keywords:** acute lymphoblastic leukemia, lineage switch, blinatumomab, minimal residual disease

## Abstract

We report incidence and deep molecular characteristics of lineage switch in 182 pediatric patients affected by B-cell precursor acute lymphoblastic leukemia (BCP-ALL), who were treated with blinatumomab. We documented six cases of lineage switch that occurred after or during blinatumomab exposure. Therefore, lineage conversion was found in 17.4% of all resistance cases (4/27) and 3.2% of relapses (2/63). Half of patients switched completely from BCP-ALL to CD19-negative acute myeloid leukemia, others retained CD19-positive B-blasts and acquired an additional CD19-negative blast population: myeloid or unclassifiable. Five patients had *KMT2A* gene rearrangements; one had *TCF3::ZNF384* translocation. The presented cases showed consistency of gene rearrangements and fusion transcripts across initially diagnosed leukemia and lineage switch. In two of six patients, the clonal architecture assessed by *IG/TR* gene rearrangements was stable, while in others, loss of clones or gain of new clones was noted. *KMT2A*-r patients demonstrated very few additional mutations, while in the *TCF3::ZNF384* case, lineage switch was accompanied by a large set of additional mutations. The immunophenotype of an existing leukemia sometimes changes via different mechanisms and with different additional molecular changes. Careful investigation of all BM compartments together with all molecular –minimal residual disease studies can lead to reliable identification of lineage switch.

## 1. Introduction

In acute lymphoblastic leukemia, the advent of immunotherapy, a form of treatment aimed at activating the immune system against tumor cells, has greatly expanded the list of effective medical options. Primarily, immunotherapy has been beneficial for the treatment of patients with refractory or relapsed (R/R) forms of disease. Blinatumomab is among the most studied and well-known immunotherapies approved for use in patients with R/R B-cell precursor acute lymphoblastic leukemia (BCP-ALL) [1,2,3,4]. This medication is a first-in-class bispecific T-cell engager (BiTE) that recruits and stimulates cytotoxic T cells to eliminate cells of B-lineage origin [5]. Its design allows the simultaneous targeting of CD3 (a T-cell antigen) and CD19, a molecule highly expressed on leukemic and normal B cells. The safety and efficacy of blinatumomab in children and adults with R/R BCP-ALL have been demonstrated in a few international clinical trials [2,3,4,6,7].

However, a notable number of patients treated with blinatumomab still experience disease recurrence [2,3]. Up to 20% of post-blinatumomab relapses are characterized by loss of the targeted B-cell antigen CD19 from leukemic cells [6,8,9]. Some CD19-negative relapses are accompanied by loss of all B-lineage antigens and acquisition of a distinct myeloid immunophenotype [10,11]. This phenomenon is known as “lineage switch” and is not limited strictly to the immunotherapy field. Lineage switch was first observed during disease relapse after standard chemotherapy [12]; however, the widespread implementation of immunotherapy has led to an increase in the number of reports [13,14,15,16,17,18]. As previously reported, any significant change in antigen expression within leukemic or normal cells becomes an obstacle for proper immunophenotypic monitoring of MRD [9]. Understanding the potential pitfalls and limitations associated with targeted therapy can facilitate early detection of relapse during disease monitoring.

In this study, we report incidence of lineage switch in pediatric patients who were treated with blinatumomab after failure of or resistance to standard chemotherapy. Herein, we present data from immunophenotypic, cytogenetic and molecular studies before and at the time of lineage switch. Given these findings, we also propose recommendations for disease monitoring in the setting of treatment with blinatumomab or other targeted therapies.

## 2. Results

### 2.1. General Treatment Information and Outcomes

We documented six cases of lineage switch that occurred after (n = 2) or during (n = 4) blinatumomab exposure in six pediatric patients affected by BCP-ALL (median age 2 years, range 14 days–12 years). Therefore, lineage conversion after selective pressure on CD19-positive cells was found in 17.4% of all resistance cases (4 of 27) and 3.2% of relapses (2 of 63). All patients were initially diagnosed with BCP-ALL according to morphological, cytochemical, and immunophenotypic evaluation of BM aspirates [19]. The results of the initial diagnostic investigations are summarized in Appendix A. In the presented cases, blinatumomab was initiated because of relapse development in three patients (pts #2, #3, and #5); failure to achieve response after first-line therapy in one patient (pt #4), and MRD reappearance during second (pt #1) or first (pt #6) remission in the remaining patients. Two patients completed one blinatumomab course (pts #2 and #3), and one patient received two complete courses (pt #1). In three patients, the blinatumomab course was interrupted by lineage switch (pts #4, #5, and #6). The overall treatment history of the patients is presented in Table 1.

A brief summary of the laboratory findings is given in Table 2.

### 2.2. Immunophenotypic Findings

Based on the criteria of The European Group for the Immunological Classification of Leukemias (EGIL) [20], primary immunophenotypic studies identified BI-ALL in five patients (pts #1, #2, #3, #4, and #6) and BIII-ALL in one patient (pt #5). CD33 and NG2 were among the most common coexpressed markers (Appendix A).

During post-blinatumomab lineage conversion, three patients (pts #1, #5, #6) switched completely from BCP-ALL to CD19-negative AML. In patient #1, 0.18% of cells were leukemic B-lineage blasts (CD19+CD10-CD45dimCD34+/-) on Day +90 after HSCT. These cells disappeared after the blinatumomab treatment course. Six months later, myeloid cells (CD19-iCD79a-iCD22-CD33+CD117+CD56+CD11c-/+) accounted for 16% of BM cells (Figure 1). FLAM block (FLAG with addition of mitoxantrone [21] but without G-CFS) with a subsequent second related HCST induced continuous complete remission. In patient #5, the previous BCP-ALL cells (CD19+iCD79a+CD10-/+CD34-CD45dim, 38.6% in BM prior to blinatumomab) had been completely replaced by myeloid leukemic cells (CD19-iCD79a-iCD22-CD33+CD117+CD34-CD14-CD11c+) by Day 27 of the blinatumomab course. These cells were resistant to all therapies attempting to decrease the leukemic burden. In patient #6, who received blinatumomab due to reappearance of B-lineage MRD three months after allo-HSCT, 5.0% of BM cells were found to be myeloid blasts (CD19-iCD79a-iCD22-CD33+CD117+CD64+CD7+) on the 22nd day of the immunotherapy course. In two weeks, their proportion increased to 75%. Due to high expression of CD38 on all leukemic cells, application of daratumumab induced MRD negativity.

Three other patients retained CD19-positive B-lineage blasts and acquired an additional CD19-negative blast population: myeloid (pts #2 and #4) or unclassifiable (pt #3). The median time between blinatumomab initiation and lineage switch was 31 days (range 20–207 days).

Two patients (pts #2 and #4) showed coexistence of two leukemic populations before the blinatumomab course. In patient #2, the main B-lineage leukemic population (15.2%) (CD19+ iCD79a+iCD22+CD10-CD45-CD34+/-CD33+/-) was accompanied by a small population of myeloid blast cells (3%) (CD33+CD19-iCD79a-iCD22-CD45+CD34-CD14-) at the time of the second post-chemotherapy relapse (Figure 2). Because of this finding, patient #2 received a high-risk chemotherapy block, which resulted in clearance of the myeloid blasts and a decrease in the B-lineage blasts to 1.5%. In one week, the proportion of the B-lineage blasts increased to 25%, but no myeloid blasts were present. Immediately after a short course of dexamethasone, patient #2 received blinatumomab. On Day 35 after blinatumomab initiation, the myeloid blasts reappeared, and their proportion exceeded that of the B-lineage blasts (CD19+) (20% versus 2%) (Figure 2). Further salvage chemotherapy was complicated by tumor lysis syndrome and acute renal failure. As a result, the patient died from treatment complications.

In patient #4, a population of myeloid blasts (CD45dimCD19-CD14+CD64+CD33+) was first observed during first-line chemotherapy. The patient showed no therapy response and retained a high MRD level. During the regular follow-up assessment, immunophenotypic evaluation revealed 4% myeloid blasts in addition to the initial B-lineage leukemic population (10.8%). The blinatumomab course was initiated four days later. On Day 22 of the course, the myeloid population increased to 83%, while the B-cell population (CD19+) was 4.3%. As all treatment options were exhausted, patient #4 was transferred to palliative care. Nine days after the lineage switch, the patient died.

Pt #3 had received blinatumomab due to BCP-ALL relapse (CD19+ iCD79a+iCD22+CD10-CD34-CD33+CD45dim) as the bridge therapy to allo-HSCT. Reappearance of CD19-positive MRD (1.3%) was noted at Day +100 after transplantation. After receiving chemotherapy including dexamethasone and cyclophosphamide, the leukemic burden increased to 22%. Simultaneously, an additional population of leukemic blasts (44%) with undifferentiated morphologic features and the absence of lineage-specific markers (including CD19) was found in the BM (Figure 3). Due to sample availability, the next time point, at which time these unclassifiable cells were completely predominant (Appendix A), was used for molecular studies. Subsequent therapy including CD19/CD22-directed CAR-T cells and second allo-HSCT failed to prevent CD19-positive BCP-ALL relapse, although an unclassifiable subpopulation of leukemic cells was successfully eliminated from the BM (Appendix A).

### 2.3. Cytogenetic Findings

According to initial cytogenetic and molecular diagnostics, five out of six patients carried *KMT2A* gene rearrangements, including t(4;11)(q21.3-q22.1;q23.3) (pts #1, #4, and #6) and t(11;19)(q23.3;p13.3) (pts #3 and #5). A DNA study with a custom *KMT2A*-targeted NGS panel revealed breakpoint junctions of forward (pts #1 and #3-#6) and reciprocal (pts #3-#6) fusion genes. *KMT2A::AFF1* and *KMT2A::MLLT1* fusion transcripts were shown to be expressed unambiguously in the corresponding RNA samples. One out of six patients had a normal karyotype with *TCF3* rearrangement with partial deletion and *ZNF384* rearrangement, as shown by FISH (pt #2). Cryptic t(12;19)(p13;p13)/*TCF3::ZNF384* was found by RNA-seq (Appendix A) and exome sequencing. Breakpoint junctions at both the DNA and RNA levels were validated with Sanger sequencing and submitted to the GenBank database; the accession numbers are summarized in Appendix A.

All patients demonstrated consistency in the main translocations across the initial and relapsed samples, as confirmed by conventional cytogenetics and FISH studies (Table 2, Appendix A). Patients #4 and #6 acquired additional chromosomal abnormalities in AML cells; however, t(4;11)(q21.3-q22.1;q23.3) was preserved (Table 2, Appendix A).

Further confirmation of the identical nature of the main translocations in the initial BCP-ALL and relapsed AML or ALAL samples of the studied patients was achieved at the molecular level. DNA breakpoint junctions in both *KMT2A*-rearranged and *TCF3*-rearranged BCP-ALL/AML pairs were Sanger-sequenced using patient-specific primers (Appendix A) and were shown to be identical (Figure 4). Additional studies of reciprocal fusion genes (where available) and fusion transcripts are summarized in Appendix A, respectively.

### 2.4. Additional Mutations Identified by Exome Sequencing

Whole-exome sequencing was performed on paired prior/post blinatumomab treatment sample pairs to determine whether the phenotype switch contributed to the acquisition of additional pathogenic mutations in genes relevant for myeloid neoplasm development. Overall, the *KMT2A*-rearranged samples demonstrated a relatively uneventful mutational landscape with a median of 6.4 nonsynonymous exonic deleterious variants in initial samples and 8.5 variants in relapse samples (Appendix A). However, we observed four distinct patterns of the distribution of additional events in this group. Pattern 1 exhibited no significant pathogenic variants, either initially or in relapse samples (pts #1 and #3). Pattern 2 (pt #5) included the appearance of a myeloid-associated pathogenic variant (*CBL* p.Y371C) in the relapse sample. Notably, this mutation had a high variant allele frequency (94.87%), which indicates loss of heterozygosity in this region typical of myeloid neoplasms. Pattern 3 included shared additional pathogenic mutations between initial and relapse samples (pt #4—*KRAS* p.G12D, pt #5—*KRAS* p.G13D). Pattern 4 was exhibited by patient #5, who developed AML relapse with the hypermutator phenotype with a total of 29 nonsynonymous deleterious exonic variants, including 2 pathogenic mutations in the *TP53* gene (*TP53* p.R89Q, *TP53* p.R16H;). Additional pathogenic mutations in lineage-switched patients are summarized in Table 2; the overall mutational landscape is presented in Appendix A.

### 2.5. Clonality Assessment by Next-Generation Sequencing

We performed comparative *BCR/TCR* clonality analysis of paired samples before and after lineage switch for all six patients (12 samples total). Approximately 17,000 genome equivalents (100 ng of gDNA) for each sample were input into the clonality assay and sequenced with an average ~600,000-read coverage per sample. We detected clonal *TCR/BCR* markers specific for leukemic clones in all 12 analyzed samples (Table 3). The identified clonal rearrangements belonged to both the *BCR* and *TCR* gene families, except *TRA*. The detected clonal rates ranged between 10% and 99% (a portion of reads covered unique rearrangements among particular *BCR/TCR* gene families). The number of markers varied from 1 to 8 per sample. We observed three patterns of leukemic clone behavior in the analyzed pairs: clone persistence (pts #2, #3, #5, and #6), clone loss (pts #1, #4, and #5), and appearance of new clones (pts #1, #3, and #4) (Table 2 and Table 3). These patterns existed individually or in different combinations, providing a relatively wide variety of clonal evolution ways for leukemic cells. However, we did not observe any associations between clonal evolution features and time from blinatumomab therapy start to lineage switch.

In four out of six patients (66.7%), several or all clonal markers were stable during blinatumomab treatment and lineage switch. This finding indicates the high probability that these markers are specific for all leukemic cells (or even for leukemic precursors) of a particular patient. They thus can be used for minimal residual disease monitoring even after lineage switch.

## 3. Discussion

While solving important clinical problems, targeted immunotherapy sets new challenges in disease monitoring and choosing consequent treatment strategies. The usage of CD19-directed therapies was shown to induce fluctuations in the expression of CD19 and other antigens on leukemic cells [8,9]. Moreover, leukemic cells can completely lose B-lineage antigens and acquire markers of other hematopoietic lineages. These two main mechanisms of tumor escape from the pressure of immunotherapy both limit possible salvage options [22] and hamper the well-established methodology of flow cytometric MRD detection [9,23,24]. Such cases were reported either after blinatumomab [25] or after CAR-T cell therapy [13,17,18,26]. In this work, we report the incidence of lineage switch in a significant cohort of children with R/R BCP-ALL after treatment with blinatumomab. Moreover, various patterns of immunophenotypic changes and molecular behavior are described. Although the study group was rather heterogeneous (by age, disease status, number of blinatumomab cycles), we consider these data representative, as lineage conversion could be the mechanism of tumor escape in any group of patients with BCP-ALL after immunotherapy.

Regardless of the applied therapeutic agent, the majority of lineage conversions are characterized by the presence of *KMT2A* gene rearrangements [27]. Among the patients presented in this study, only one patient did not carry a *KMT2A* rearrangement but was found to have t(12;19)(p13;p13)/*TCF3::ZNF384*. This genetic aberration has been frequently observed in acute leukemia with a mixed B/myeloid immunophenotype [18]. Also in described patient, the additional very small myeloid subpopulation was visible during the whole course of relapse treatment prior to lineage switch. Cases of lineage conversion in patients harboring *TCF3::ZNF384* translocation were recently reported [18]. Our findings as well as published data [18,28,29] indicate that lineage switch is not restricted to *KMT2A*-rearranged cases and can be anticipated in ALL with *ZNF384* rearrangement.

The immunophenotypic presentation of the described cases was different. Along with the generally typical substitution of BCP-ALL by myeloid leukemia (n = 3), in two cases, pre-existing myeloid cells benefited from the selective pressure targeting the lymphoid population. The remaining case demonstrated the appearance of unclassifiable leukemic cells with a completely unspecific immunophenotype but retained genetic characteristics. This case included the most complicated confirmation of lineage switch because of the coexistence of BCP-ALL cells and cells of an unexpected and completely unclassifiable immunophenotype. In contrast, two cases with a pre-existing myeloid subpopulation included easier recognition of lineage switch, as the presence of such a population prior to CD19 targeting was an obvious sign of a possible lineage switch. Cases of complete BCP-ALL substitution by AML are also foreseeable, as they have occurred in patients with *KMT2A* rearrangements, but their timely identification required careful investigation of all main BM compartments during MFC-MRD analysis, as recommended previously [9]. Half of the described cases included recognition of the lineage switch before completion of the blinatumomab course; hence, it was impossible to predict the lineage switch by MRD studies. Regardless, the detection of myeloid cells at the MRD level would likely be very difficult even if the B-lineage MRD antibody set is expanded with additional markers.

The leukemic clone behavior assessed by investigation of *IG/TR* gene rearrangements was also different in the studied cases, representing three different patterns, which were sometimes found in the same cases. Only in two of six patients was the clonal architecture stable, while in others, loss of clones or gain of new clones was noted. In terms of overall clonal fluctuations, four out of six analyzed samples had at least two stable *IG/TR* targets for MRD monitoring. Furthermore, two samples completely lacked appropriate IG/TR targets for MRD monitoring. Thus, the initial presence of markers specific for dominant leukemic clones in each particular case does not guarantee the reliability of MRD monitoring due to the rather high probability of IG/TR marker disappearance at lineage switch. The mutational profile of the described cases was also different. As expected [30], *KMT2A*-r patients demonstrated very few additional mutations, with only one out of five patients acquiring pathogenic events in AML relapse. In contrast, in children with t(12;19)(p13;p13)/*TCF3::ZNF384*, a change in leukemic subpopulation distribution was accompanied by a large set of additional mutations. Our findings on additional mutations are in line with *KRAS* being frequently mutated in primary BCP-ALL as well as in relapse and *TP53* being enriched in relapse [31,32]. Among molecular markers, only gene fusions and their transcripts displayed absolute stability during lineage conversion. This stability enabled us to confirm that all the “new” leukemic subpopulations were indeed new incarnations of the same tumor with changed immunophenotype and several changed molecular characteristics. Moreover, these fusions could serve as stable targets for MRD monitoring [33].

Indeed, the two-thirds of lineage switches appeared in the form of tumor refractoriness to CD19 targeting. Only two of the six cases included relapse after achievement of complete remission. Both cases with pre-existing myeloid tumor populations were resistant to blinatumomab. Thus, in patients with an existing myeloid leukemia population, regardless of the presence of *KMT2A*-r, the use of blinatumomab can lead to growth of this myeloid subpopulation. Indeed, one can classify such cases as MPAL due to presence of cells with immunophenotypic features of both lymphoid and myeloid lineages. Nevertheless, small visible (on the MRD level) myeloid tumor population on the background of huge BM infiltration by B-lineage blasts is never considered as the obstacle for the blinatumomab application [34,35,36]. Such selection of pre-existing myeloid tumor population is also considered as the lineage switch [37]. Moreover, even for B/Myeloid MPAL, targeting of CD19 does not always lead to the selection of myeloid population [34,35,36], especially if myeloid-type therapy is used [34].

Finally, two main recommendations for MRD monitoring after CD19-directed treatment could be stated. First, careful investigation of all BM compartments prior to and after course of blinatumomab, especially in patients with *KMT2A* rearrangements and *ZNF384* fusions, can result in timely diagnosis of lineage switch. Suspicious myeloid cells could be found according to expression of CD45, which is always the part of MRD-oriented panels in BCP-ALL and light-scatter properties. In a part of the samples, evaluation of unusual myeloid cells could be performed using CD24, antigen, which is recommended for use in MFC-MRD monitoring after CD19-directed therapy as the possible substitution for CD19 [38]. Thereafter, such cells should be investigated with more detailed immunophenotyping or other ways of confirmation of their tumor origin [39]. On the other hand, one of the main results of the current work is demonstration of stability of fusion genes and their transcripts as the possible MRD targets, in contrast to *IG/TR* rearrangements and mutation patterns. Therefore, high levels of MRD assessed by detection of fusions with the absence of the huge amount of B-lineage blasts by MFC have to be considered as the possible indication of probable lineage conversion.

Despite detailed description of immunophenotypic and molecular changes, it is still difficult to make a definitive conclusion of the prognostic value of lineage conversion. Of course, cases of resistance to blinatumomab via lineage switch have grim prognosis, although for patients for whom completion of CD19-targeting did not result in immediate switch, subsequent myeloid-style treatment could be the way for relapse prevention even for children with pre-existing myeloid tumor subpopulation [34].

## 4. Materials and Methods

### 4.1. Patients

Between January 2016 and September 2021, a total of 182 children (0–18 years old) with BCP-ALL were treated with blinatumomab in four medical institutions. Twenty-three patients were resistant to immunotherapy, while 67 experienced relapse following achievement of complete remission. All patients or their legal guardians signed informed consent forms.

### 4.2. Blinatumomab Schedule

A blinatumomab course of continuous intravenous infusion at doses of 5 µg/m^2^/day (Days 1–7) and 15 µg/m^2^/day (Days 8–28) was administered for four weeks. The doses were escalated to fixed values of 9 µg/day (Days 1–7) and 28 µg/day (Days 8–28) for elderly patients whose weight was greater than or equal to 45 kg.

### 4.3. Immunophenotyping

Bone marrow (BM) samples were studied using flow cytometry at the time of diagnosis and relapse and were assessed for MRD presence at follow-up time points with respect to the treatment protocols. The diagnostic panels of fluorochrome-conjugated monoclonal antibodies were based on Moscow–Berlin-group diagnostic standards [40]. The antibody panels for MRD monitoring were reported previously [41]. The flow cytometry data were collected on FACSCanto II (Becton Dickinson, San Jose, CA, USA), Navios (Beckman Coulter, Indianapolis, IN, USA) and CytoFLEX (Beckman Coulter) flow cytometers and then analyzed using Kaluza 2.1 software (Beckman Coulter). EuroFlow guidelines for machine performance monitoring were used [42]. Cytometer Setup and Tracking Beads (BD), Flow-Check Pro Fluorospheres (Beckman Coulter) and CytoFLEX Daily QC Fluorospheres (Beckman Coulter) were used for daily cytometer optimization.

### 4.4. Evaluation of Breakpoint Junctions in Fusion Genes and Fusion Transcripts

BM aspirates obtained at diagnosis and relapse were cultured overnight without mitogenic stimulation and processed as previously described [43]. GTG-stained karyotypes were recorded as per the International System for Human Cytogenomic Nomenclature 2020 [44]. FISH analysis with corresponding probes was performed at diagnosis and relapse according to the manufacturer’s instructions; at least 200 interphase nuclei were analyzed in each case. The probes used were as follows: XL *KMT2A* Plus break-apart (MetaSystems GMBH, Altlussheim, Germany), Kreatech ON *KMT2A/AFF1* t(4;11) fusion or Kreatech ON *KMT2A/MLLT1* t(11;19) fusion (Leica Microsystems B.V., Amsterdam, The Netherlands) for *KMT2A*-rearranged samples, and CytoTest *E2A* break-apart (CytoTest, Rockville, MD, USA) and Cytocell *ZNF384* break-apart (Cytocell, Milton, Cambridge, UK) for samples without *KMT2A* rearrangement. Percentage of rearranged cells is summarized in Appendix A.

Total DNA and RNA were simultaneously extracted from BM aspirates using the InnuPrep DNA/RNA Mini Kit (Analytik Jena AG, Jena, Germany). DNA breakpoint junctions were defined for *KMT2A*-rearranged samples with a targeted enrichment panel for the whole *KMT2A* gene [45], followed by next-generation sequencing (NGS) on the Illumina MiSeq platform (Illumina, San Diego, CA, USA) and validated by Sanger sequencing for all diagnosis and relapse samples. The expression of the corresponding fusion transcripts was demonstrated by RT–PCR on total RNA for all diagnosis and relapse samples. Samples lacking *KMT2A* gene rearrangement were subjected to transcriptome analysis. RNA for the transcriptome analysis was extracted using TRIzol reagent (Thermo Fisher Scientific, Waltham, MA, USA).

The RNA-seq library was prepared using the NEBNext Ultra II Directional kit (NEB, Ipswich, MA, USA); sequencing was performed on the Illumina NextSeq platform (Illumina). NGS data were aligned to the reference genome using the STAR algorithm [46]. Fusion transcripts were found using the Arriba algorithm [47] and validated by Sanger sequencing. The direct genomic DNA PCR and RT–PCR primers are listed in Appendix A.

### 4.5. Exome Sequencing

NGS libraries were prepared from 300 to 600 ng of genomic DNA using the MGIEasy Universal DNA Library Prep Set (MGI Tech, Shenzhen Shi, China), according to the manufacturer’s protocol. DNA was fragmented to an average fragment length of 250 bp on a Covaris S-220 device (Covaris, Woburn, MA, USA). DNA enrichment was carried out on preliminary pooled libraries [48] using SureSelect Human All Exon v7 probes, recognizing the complete human genome (Agilent, Santa Clara, CA, USA). The libraries were circularized and sequenced in paired-end mode on the MGISEQ-2000 platform using the DNBSEQ-G400RS High-throughput Sequencing Set PE100 according to the manufacturer’s protocol (MGI Tech), with an average coverage of 100x. The FastQ files were generated using the manufacturer’s zebracallV2 software (MGI Tech, Shenzhen Shi, China).

Raw data quality control was conducted with FastQC (v0.11.9) [49]. Raw reads were filtered, and adapters were trimmed using fastp (v0.19.4) [50]. After quality control, the reads were mapped to the hg38 reference primary assembly genome (Release 38) from GENCODE [51] by BWA (v0.7.17) [52]. SAM files were converted to BAM files and sorted by coordinates with SAMtools (v1.13) [53]. Duplicated reads were marked with Picard (v2.26.1) (Picard Tools, Broad Institute; available from http://broadinstitute.github.io/picard, accessed on 10 February 2022). Variant calling was performed with the Genome Analysis Tool Kit Mutect2 in tumor-only mode (v4.2.2.0) following GATK’s best practice [54]. The resulting mutations were annotated with Annovar (v2019Oct24) [55] and VEP (v104.3) (EMBL’s European Bioinformatics Institute, Hinxton, Cambridgeshire, UK) [56].

### 4.6. Detection of Clonal Rearrangements by Next-Generation Sequencing

Clonality assessment was performed using high-throughput sequencing of *TCR/BCR* genes with rearrangements. Sequencing libraries were constructed as previously described [57] using two-step PCR and genomic DNA as input material. Target enrichment was performed in the first PCR step with a multiplex primer set for the V, D, and J segments of all rearranged *TCR* and *BCR* genes rearrangements: *TRA*, *TRB*, *TRG*, *TRD*, *IGH*, *IGK*, and *IGL* (MiLaboratory LLC., Sunnyvale, CA, USA-, https://milaboratories.com/kits accessed on 10 February 2022). Each PCR (25 μL) contained primer mix (MiLaboratory), 100 ng of genomic DNA, six units of HS Taq polymerase in 1X Turbo buffer, and dNTPs (0.125 μmol/L each) (all Evrogen, Russia). The amplification profile was as follows: 94 °C for 3 min (initial denaturation), 94 °C for 20 s, 56 °C for 90 s, and 72 °C for 40 s for ten cycles, followed by 94 °C for 20 s and 72 °C for 90 s for 15 cycles with a ramp rate of 0.5 °C/s for all stages. The obtained amplicons were cleaned up using one volume of magnetic AmPure XP beads (Beckman Coulter), according to the manufacturer’s protocol. Second, PCR with UDI primers (Illumina) was used to attach sample indices and adapters to the libraries. The final libraries were cleaned up as described above, pooled together, and sequenced on a MiSeq instrument (paired-end 150 nt reads). *TCR* and *BCR* rearrangement repertoires were extracted from sequencing data using modified MiXCR software [58] and converted into VDJtools software format [59]. Amplification errors were corrected using the Correct function from the VDJtools package. Multiplex-specific quantitative biases were corrected using iROAR software [60]. A frequency of 5% was used as a cutoff to identify rearrangements specific for leukemic clones.

## 5. Conclusions

Understanding the potential risk factors for lineage switch and developing a strategy for monitoring patients at risk are of high importance. In the presented cases, the immunophenotype of an existing leukemia sometimes changed via different mechanisms and with different additional molecular changes. In addition, different patients, not only those carrying *KMT2A* rearrangements, can undergo such lineage conversion under the selective pressure of blinatumomab. We can conclude that careful investigation of all BM compartments (at least according to CD45 expression and light-scatter properties) in addition to conventional B-lineage MRD investigation with precise inspection of all suspicious cell populations can lead to reliable, timely identification of lineage switch. Moreover, the results of all types of molecular MRD studies, especially the detection of gene fusions, can be valuable for flow cytometry identification of patients who are highly likely to undergo lineage conversion.

## Figures and Tables

**Figure 1 ijms-23-04019-f001:**
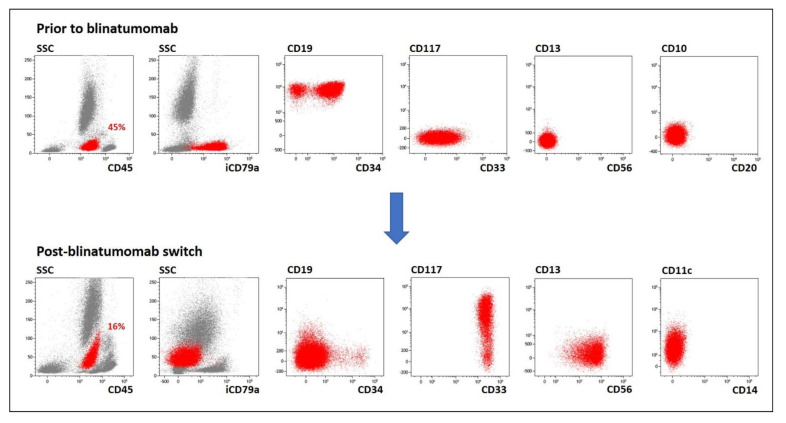
Example case of complete substitution of BCP-ALL by AML (pt #1). Tumor cells are shown in red; other bone marrow cells are gray. SSC—side-scatter, i—intracellular expression.

**Figure 2 ijms-23-04019-f002:**
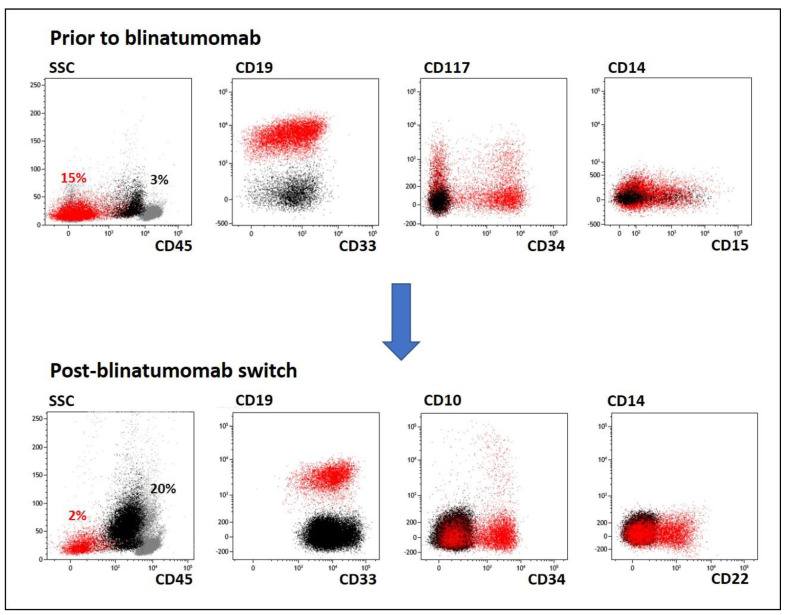
Example case of selection of a pre-existing myeloid population (pt #2). BCP-ALL cells are shown in red; myeloid cells are painted black; other bone marrow cells are gray. SSC—side-scatter.

**Figure 3 ijms-23-04019-f003:**
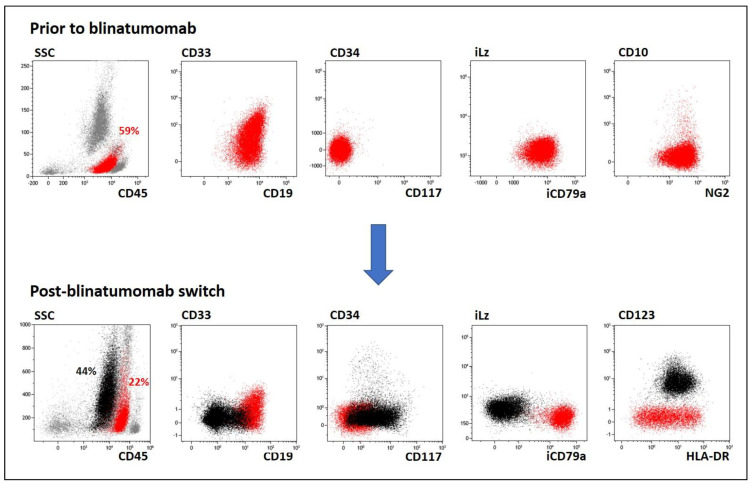
Case of acquisition of an additional unclassifiable leukemic population (black) on the background of still-existing BCP-ALL cells (red) (pt #3). Other bone marrow cells are gray. SSC—side-scatter.

**Figure 4 ijms-23-04019-f004:**
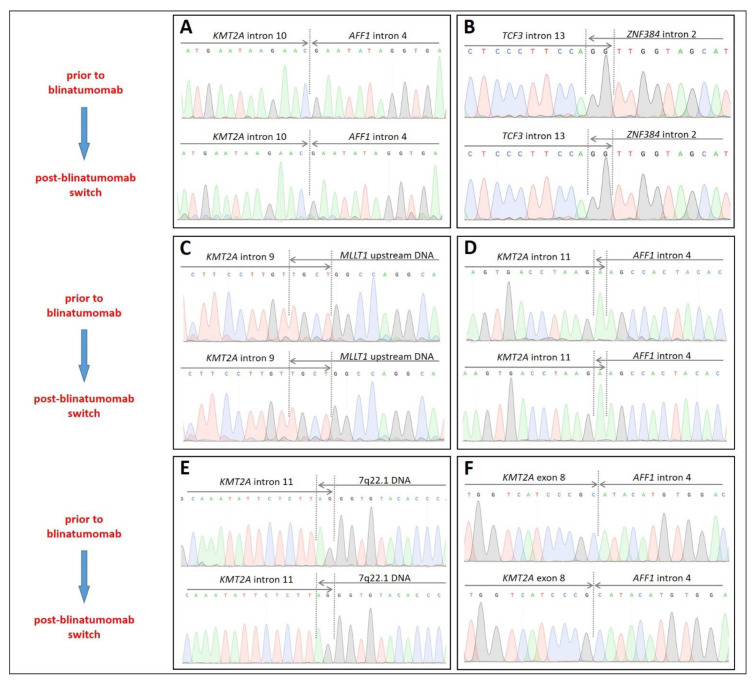
Molecular studies in lineage-switched patients demonstrating consistency of the main chromosomal translocations. (**A**)—*KMT2A::AFF1* fusion gene in pt #1; (**B**)—*TCF3::ZNF384* fusion gene in pt #2; (**C**)—*KMT2A::MLLT1* fusion gene in pt #3; (**D**)—*KMT2A::AFF1* fusion gene in pt #4; (**E**)—*KMT2A-*7q22.1 part of a fusion gene in #5 (complex *KMT2A-*7q22.1*-MLLT1* upstream DNA fusion gene is depicted in Appendix A); (**F**)—*KMT2A::AFF1* fusion gene in pt #6.

**Table 1 ijms-23-04019-t001:** Treatment and response history.

Patient #	Initial Diagnosis [Age] → Diagnosis at Lineage Switch [Age]	Treatment before Blinatumomab [Result]	Blinatumomab Course [Result]	Post-Switch Treatment [Result]	Time from Lineage Switch to Death
#1	BCP-ALL [12 years] → AML [13 years]	ALL-MB 2015 [MRD(+) CR] Allo-HSCT [1st relapse at 3 months]	Blinatumomab x2 [MRD(-) CR, then 2nd relapse and lineage switch at 4 months]	FLAM block → allo-HSCT [MRD(-) CR]	Alive in CR
#2	BCP-ALL [3 years] → ALAL (B + myeloid) [10 years]	ALL-BFM 90 [CR, then 1st relapse at 5 years] ALL-REZ-MB 2014 [CR, then 2nd relapse at 1 year] HR3 block [ALL-REZ-MB 2014]	Blinatumomab [lineage switch at 35 days]	Cytarabine + clofarabine [no response] → palliative care	39 days
#3	BCP-ALL [10 years] → ALAL (B + unclassifiable) [11 years]	ALL-MB 2015 [MRD(-) CR, then 1st relapse at 9 months]	Blinatumomab → allo-HSCT [MRD(-) CR, then 2nd relapse and lineage switch at 3.8 months]	Dexamethasone + cyclophosphamide → hAM24 → allo-HSCT + CAR-T cells [MRD(-) CR, then 3rd relapse at 2 months] → palliative care	8 months
#4	BCP-ALL [14 days] → ALAL (B + myeloid) [6 months]	MLL Baby [no response]	Blinatumomab [no response, lineage switch at 22 days]	Palliative care	9 days
#5	BCP-ALL [11 months] → AML [2 years]	MLL Baby [MRD(-) CR, then 1st relapse at 9 months] ALL-REZ-MB 2014 [MRD(+) CR]	Blinatumomab [lineage switch at 27 days]	Dexamethasone + etoposide + daunorubicin [no response] → palliative care	42 days
#6	BCP-ALL [6 months] → AML [6 years]	MLL Baby induction + Interfant 2006 [MRD(-) CR, then 1st relapse at 5 years] ALL-MB 2015 → allo-HSCT [CR, then MRD-reappearance]	Blinatumomab [lineage switch at 20 days]	Daratumumab [unknown]	LFU

BCP-ALL, B-cell precursor acute leukemia; AML, acute myeloid leukemia; ALAL, acute leukemia of ambiguous lineage; MRD, minimal residual disease; CR, complete remission; allo-HSCT, allogeneic hematopoietic stem cell transplantation; CAR-T, chimeric antigen receptor T; LFU, lost to follow-up; y, years; mo, months; d, days; N/A, not applicable.

**Table 2 ijms-23-04019-t002:** Key laboratory findings.

Patient #	Diagnosis	Immunophenotypic Markers	Karyotype	Cytogenetic Aberration	Fusion Gene Breakpoint Junction	Fusion Transcript Breakpoint Junction	Additional Pathogenic Mutations	Feature of Clonal Evolution
1	BCP-ALL	CD19+iCD79a+CD10-CD45+CD34+/-	No metaphases	t(4;11)(q21;q23)/*KMT2A::AFF1*	Identical	Identical	-	Clone loss + new clones
AML		CD19- iCD79a-CD33+CD117+CD56+CD11c-/+	47,XX,t(4;11)(q21.3-q22.1;q23.3),+6,der(15)add(p13) [10]				-	
2	BCP-ALL	CD19+iCD79a+CD10-CD45+	46,XX [8]	t(12;19)(p13;p13)/*TCF3::ZNF384*	Identical	Identical	-	Leukemic clone persistence
ALAL (B + myeloid)		(1) CD19+iCD79a+CD10-CD34+CD33+CD45- (2) CD19-CD10-CD34+CD33+CD45+	No metaphases				*TP53* p.R89Q *TP53* p.R16H	
3	BCP-ALL	CD19+ iCD79a+iCD22+CD10-CD34-CD33+CD45+	47,XX,+6,t(11;19)(q23.3;p13.3) [17]/46,XX [3]	t(11;19)(q23;p13)/*KMT2A::MLLT1*	Identical	Identical	-	Leukemic clone persistence + new subclone
ALAL (B + unclassifiable)		(1) CD19+CD33-CD34-CD45+ (2) CD19-CD33-CD34-CD117+/-CD45+CD123+	No metaphases				-	
4	BCP-ALL	CD19+iCD79a+CD22+CD10-CD45+	46,XX,t(4;11)(q21.3-q22.1;q23.3),der(13)c [11]	t(4;11)(q21;q23)/*KMT2A::AFF1*	Identical	Identical	*KRAS* p.G12D	Clone loss + new clones
ALAL (B + myeloid)		(1) CD19+CD10-CD45+ (2) CD19-CD14+CD64+CD33+	47,XX,t(4;11)(q21.3-q22.1;q23.3),+der(4)t(4;11)(q21.3-q22.1;q23.3)der(13)c [11]				*KRAS* p.G12D	
5	BCP-ALL	CD19+iCD79a+CD22+CD10-/+CD34-CD45+	No metaphases	t(11;19)(q23;p13)/*KMT2A::MLLT1*	Identical	Identical	-	Leukemic clone persistence
AML		CD19-CD33+CD117+CD34-CD14-CD11c+	No metaphases				*CBL* p.Y371C	
6	BCP-ALL	CD19+iCD79a+CD10-CD45+	47,XX,t(4;11)(q21.3-q22.1;q23.3),+22 [6]	t(4;11)(q21;q23)/*KMT2A::AFF1*	Identical	Identical	*NRAS* p.G13D	Leukemic clone persistence
AML		CD19-iCD79a-iCD22-CD33+CD117+CD64+CD7+	56,XX,+X,+der(4)t(4;11)(q21.3-q22.1;q23.3),+der(4)t(4;11)(q21.3-q22.1;q23.3), +der(6),+der(6),+7,+8,der(11)t(4;11)(q21.3-q22.1;q23.3),+13,+17,+22 [14]/46,XX [1]				*NRAS* p.G13D	

i—intracellular expression.

**Table 3 ijms-23-04019-t003:** Stability of BCR/TCR markers specific for leukemic clones during blinatumomab treatment.

Patient #		Number of Leukemia-Related IG/TCR Rearrangements														Number of Stable IG/TCR Rearrangements	Feature of Clonal Evolution
		Before and after Blinatumomab							Common **								
		IGH *	IGK	IGL	TRA	TRB	TRG	TRD	IGH	IGK	IGL	TRA	TRB	TRG	TRD		
1	prior to blinatumomab	3	0	0	0	0	0	1	0	0	0	0	0	0	0	0	Clone loss *** + new clones ****
	switch after blinatumomab	2	0	0	0	2	1	2									
2	prior to blinatumomab	1	0	0	0	1	0	0	1	0	0	0	1	0	0	2	Leukemic clone persistence *****
	switch after blinatumomab	1	0	0	0	1	0	0									
3	prior to blinatumomab	2	2	1	0	0	0	1	2	2	1	0	0	0	1	6	Leukemic clone persistence + new subclone
	switch after blinatumomab	2	2	1	0	1	1	1									
4	prior to blinatumomab	1	0	0	0	0	0	0	0	0	0	0	0	0	0	0	Clone loss + new clones
	switch after blinatumomab	0	0	1	0	0	0	0									
5	prior to blinatumomab	2	0	0	0	0	0	1	2	0	0	0	0	0	0	2	Leukemic clone persistence
	switch after blinatumomab	2	0	0	0	0	0	0									
6	prior to blinatumomab	1	1	0	0	0	0	0	1	1	0	0	0	0	0	2	Leukemic clone persistence
	switch after blinatumomab	1	1	0	0	0	0	0									

* BCR/TCR gene family. ** Number of leukemia-related rearrangements common to a leukemic sample before and after blinatumomab treatment. *** Disappearance of leukemic clones with the initial BCR/TCR markers after blinatumomab treatment. **** Emergence of leukemic clones with new BCR/TCR markers after blinatumomab treatment. ***** Retention of clones with initial BCR/TCR markers after blinatumomab treatment.

## Data Availability

All the data and material are available upon reasonable request to uralcytometry@gmail.com.

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
