# Peer review of "Lineage Conversion in Pediatric B-Cell Precursor Acute Leukemia under Blinatumomab Therapy"

_ijms, 2022, doi:10.3390/ijms23074019_

Round 1

Reviewer 1 Report

The paper provides the reader incidence and a description of immunophenotype, cytogenetics, and molecular mutations of lineage switch cases occurring in six BCP-ALL pediatric patients after blinatumomab treatment.

The argument is of interest since immunotherapy is gaining more and more importance in treating refractory or relapsed BCP-ALL. Regardless, the manuscript requires the following Major revision before publication:

  • The described case series is widely heterogeneous: it includes patients of different ages (infants, children, adolescents) who received blinatumomab after different chemotherapy lines and regimens; the time-interval between blinatumomab administration and lineage switch was extremely wide. Additionally, no data regarding the other 176 BCP-ALL without lineage switch are available. Therefore, the presented data do not respond to the purpose of the study presented at the end of the Introduction section (we also propose recommendations for disease monitoring in the setting of treatment with blinatumomab or other targeted therapies) and do not adequately sustain the conclusions of the manuscript.
  • Two out of six patients at diagnosis already presented a myeloid subpopulation co-existing with the B-lineage lymphoblasts, classifying these diseases as mixed-phenotype acute leukemias according to WHO classification. In these cases, the disappearance of CD19 positive blast population should not be interpreted as a lineage switch but the result of immunotherapy selective pressure.
  • The authors should provide the immunophenotypic definition of CD19-negative AML, first excluding the positivity of the other B-lineage antigens cyCD79a, cyCD22, and CD10.

Minor revisions:

  • Abstract, lines 1-2: please change 182 pediatric patients to 182 pediatric patients affected by B-cell precursor acute lymphoblastic leukemia (BCP-ALL)
  • Abstract, line 5: please provide the extended version of the acronym AML
  • Introduction, page 1, line 1: please change acute leukemia to acute lymphoblastic leukemia
  • Introduction, page 2, line 3: B-cell precursor acute leukemia may be changed into B-cell precursor acute lymphoblastic leukemia to explain the acronym BCP-ALL
  • Introduction, page 2, line 3: please add the reference: JAMA 2021 Mar 2;325(9):843-854.doi: 10.1001/jama.2021.0987. Effect of Blinatumomab vs Chemotherapy on Event-Free Survival Among Children With High-risk First-Relapse B-Cell Acute Lymphoblastic Leukemia: A Randomized Clinical Trial. Franco Locatelli, et al
  • Introduction, page 2, line 8 (a few international clinical trials): please provide references
  • Results, page 2, line 2: please add “affected by BCP-ALL” after pediatric patients
  • Results, from page 2, line 9 to page 3, line 3: this section is hard to read. Please summarize it, including only the most relevant information (also considering that all the described data are already reported in the Tables.
  • Results, page 3, line 9: please change Table S2 to Table S1
  • Results, page 3, line 15: please define of FLAM block and an appropriate reference
  • Results, page 3: the usefulness of the reported in-brackets antigens is not clear (e.g. CD45, CD34). I would rather use B-lineage and myeloid-lineage specific markers to highlight the difference in the phenotype before and after treatment
  • Results, page 3, line 23: patient 6 received daratumumab. Please add data on CD38 expression.
  • Results, page 3, line 41: please specify which antigens allow to discriminate these cells from normal monocytes and define them as myeloid blasts
  • Table 2: Please provide information about other B-lineage markers other than CD19 in BCP-ALL (necessary since most of the patients are CD10-)
  • Apart Ref 16, for ZFN384, authors should consider: Leukemia. 2021 Nov;35(11):3272-3277.

doi: 10.1038/s41375-021-01199-0. Epub 2021 Mar 10. Clinical characteristics and outcomes of B-ALL with ZNF384 rearrangements: a retrospective analysis by the Ponte di Legno Childhood ALL Working Group

Shinsuke Hirabayashi  1 , Ellie R Butler  2 , Kentaro Ohki  3 , Nobutaka Kiyokawa  3 , Anke K Bergmann  4 , Anja Möricke  5 , Judith M Boer  6   7 , Hélène Cavé  8 , Giovanni Cazzaniga  9 , Allen Eng Juh Yeoh  10 , Masashi Sanada  11 , Toshihiko Imamura  12 , Hiroto Inaba  13 , Charles Mullighan  13 , Mignon L Loh  14 , Ulrika Norén-Nyström  15 , Agata Pastorczak  16 , Lee-Yung Shih  17 , Marketa Zaliova  18 , Ching-Hon Pui  13 , Oskar A Haas  19 , Christine J Harrison  2 , Anthony V Moorman  2 , Atsushi Manabe  20

16) Ref 7 is no more “Online ahead of print” but already published:  Immunophenotypic changes of leukemic blasts in children with relapsed/refractory B-cell precursor acute lymphoblastic leukemia, who have been treated with Blinatumomab. Mikhailova E, Gluhanyuk E, Illarionova O, Zerkalenkova E, Kashpor S, Miakova N, Diakonova Y, Olshanskaya Y, Shelikhova L, Novichkova G, Maschan M, Maschan A, Popov A.Haematologica. 2021 Jul 1;106(7):2009-2012. doi: 10.3324/haematol.2019.241596.

Author Response

Referee #1 (Comments to the Author):

The paper provides the reader incidence and a description of immunophenotype, cytogenetics, and molecular mutations of lineage switch cases occurring in six BCP-ALL pediatric patients after blinatumomab treatment.

The argument is of interest since immunotherapy is gaining more and more importance in treating refractory or relapsed BCP-ALL. Regardless, the manuscript requires the following Major revision before publication:

Response to Referee 1

Comment 1.

The described case series is widely heterogeneous: it includes patients of different ages (infants, children, adolescents) who received blinatumomab after different chemotherapy lines and regimens; the time-interval between blinatumomab administration and lineage switch was extremely wide. Additionally, no data regarding the other 176 BCP-ALL without lineage switch are available. Therefore, the presented data do not respond to the purpose of the study presented at the end of the Introduction section (we also propose recommendations for disease monitoring in the setting of treatment with blinatumomab or other targeted therapies) and do not adequately sustain the conclusions of the manuscript.

Response.

We agree that the described cohort is heterogeneous. Nevertheless, we think that the data is still representative. Lineage conversion can occur irrespective to age, or disease burden, or threatment stage, or  number of blinatumomab cycles. It is mainly limited to the cases with peculiar genetics, and this is shown in our study. The idea of the study was not to show the structure of relapses in any protocol including blinatumomab, but to show incidence and molecular patterns of lineage switch that can occur in case of blinatumomab administration in all possible situations. From our data one can see that lineage conversion could develope in different ways, as resistance or relapse, not only in KMT2A-r patients, with additional mutations and without them etc. So we can say that this phenomenon could be the serious problem for immunotherapy of BCP-ALL and for MRD monitoring after such type of treatment irrespective to precise treatment protocol in which blinatumomab is included. We have added this statement to the discussion as well (lines 246-249). We have not planed to talk about remaining patients and their treatment results because this is the area of another study. Moreover, they were treated in different ways and it is not easy to put such information to the article containing so detailed description of six switched cases. We wanted just to focus attention on the possible types of lineage switch and molecular features of the lineage conversion. We still think that we have concentrated the largest number of switch-lineage cases after blinatumomab treatment and tried to show cytometric and molecular patterns of this phenomenon. We think also that two very important points were highlighted in the field of MRD monitoring after CD19 targeting. First, the consistency of fusion genes and their transcripts, in contrast to more widespread MRD targets (IG/TR rearrangements) is enough solid for proposal of use these molecular markets as the most informative MRD targets. Second, necessity of the careful investigation of all BM compartments prior to and after blinatumomab exposure, especially in the case of high FG- or FGT-MRD results, is also very valuable recommendation. Indeed, it is too difficult to open cytometric discussion regarding antigens that should be added to the antibodies panels for the reliable identification of abnormal myeloid cells, in such paper that concentrates a lot of laboratory information. Nevertheless, we have added some general consideration regarding appropriate markers as well (lines 309-322).

Comment 2. 

Two out of six patients at diagnosis already presented a myeloid subpopulation co-existing with the B-lineage lymphoblasts, classifying these diseases as mixed-phenotype acute leukemias according to WHO classification. In these cases, the disappearance of CD19 positive blast population should not be interpreted as a lineage switch but the result of immunotherapy selective pressure.

Response.

We thank the reviewer for this comment. Unfortunately, we cannot fully agree with this statement. Very small visible myeloid population, which is indeed on the MRD level in both described cases, not always leads to classifying the case as MPAL. As we know now from molecular studies, all parts of such tumors are really parts of the same tumor, although very changeable, especially in case of KMT2A-rearrangement (ZNF384 fusion as well). Moreover, there is still no strict definition of the threshold for second population proportion to classify case as MPAL. That is why patients who are similar to described here, are mainly considered to be treated with lymphoid-style therapy and classified mainly as BCP-ALL with some myeloid “tail”. Moreover, even “classical” MPALs with comparable sizes of lymphoid and myeloid populations can be successfully treated with blinatumomab, especially in combination with myeloid-type consolidation. Finally, selection of the pre-existed myeloid subpopulation under the pressure of immunotherapy is one of the known way of lineage switch. We agree that it is necessary to discuss this, hence we have added these points with appropriate references to the Discussion section (lines 301-308)

Comment 3. 

The authors should provide the immunophenotypic definition of CD19-negative AML, first excluding the positivity of the other B-lineage antigens cyCD79a, cyCD22, and CD10

Response.

We thank the reviewer for this comment. We completely agree that it is necessary to show not only CD19-negativity, but also absence of other specific B-lineage antigens. Available information was added to the text and tables.

Comment 4. 

Abstract, lines 1-2: please change 182 pediatric patients to 182 pediatric patients affected by B-cell precursor acute lymphoblastic leukemia (BCP-ALL)

Response.

Corrected

Comment 5. 

Abstract, line 5: please provide the extended version of the acronym AML

Response.

Corrected

Comment 6. 

Introduction, page 1, line 1: please change acute leukemia to acute lymphoblastic leukemia

Response.

Corrected.

Comment 7. 

Introduction, page 2, line 3: B-cell precursor acute leukemia may be changed into B-cell precursor acute lymphoblastic leukemia to explain the acronym BCP-ALL

Response.

Corrected

Comment 8. 

Introduction, page 2, line 3: please add the reference: JAMA 2021 Mar 2;325(9):843-854.doi: 10.1001/jama.2021.0987. Effect of Blinatumomab vs Chemotherapy on Event-Free Survival Among Children With High-risk First-Relapse B-Cell Acute Lymphoblastic Leukemia: A Randomized Clinical Trial. Franco Locatelli, et al

Response.

Added

Comment 9. 

Introduction, page 2, line 8 (a few international clinical trials): please provide references

Response.

Added

Comment 10. 

Results, page 2, line 2: please add “affected by BCP-ALL” after pediatric patients

Response.

Corrected.

Comment 11. 

Results, from page 2, line 9 to page 3, line 3: this section is hard to read. Please summarize it, including only the most relevant information (also considering that all the described data are already reported in the Tables.

Response.

We have shortened this part of the manuscript (lines 102-112)and tried to say only relevant points that are not shown in tables.

Comment 12. 

Results, page 3, line 9: please change Table S2 to Table S1

Response.

Corrected

Comment 13. 

Results, page 3, line 15: please define of FLAM block and an appropriate reference

Response.

Actually, this is just a Mito-FLAG, although without G-CSF. This explanation is given in text (line 125) with the reference to FLAG plus Mitoxantrone design.

Comment 14. 

Results, page 3: the usefulness of the reported in-brackets antigens is not clear (e.g. CD45, CD34). I would rather use B-lineage and myeloid-lineage specific markers to highlight the difference in the phenotype before and after treatment

Response.

We think that CD45 expression is important part of immunophenotype description as well as immaturity of blasts mainly described by CD34 expression. Nevertheless, we have added more data of myeloid and lymphoid antigens expression to make differences more visible. Unfortunately, it is not easy to compare antigens expression in text, so part of the information is shown in figures 1-3.

Comment 15. 

Results, page 3, line 23: patient 6 received daratumumab. Please add data on CD38 expression

Response.

We have added this statement in the text (lines 134-135)

Comment 16. 

Results, page 3, line 41: please specify which antigens allow to discriminate these cells from normal monocytes and define them as myeloid blasts

Response.

Indeed it is not very easy sometimes to prove that huge amount of monocytic cells (mature and maturing) are parts of the tumor. In some cases it is necessary to sort them in order to find similar clonal structure with the B-lineage blasts. Fortunately, in described case these cells expressed CD45 on lower level as compared to mature normal monocytes. This is also added to the text (line 152)

Comment 17. 

Table 2: Please provide information about other B-lineage markers other than CD19 in BCP-ALL (necessary since most of the patients are CD10-)

Response.

Available data was added.

Comment 18. 

Apart Ref 16, for ZFN384, authors should consider: Leukemia. 2021 Nov;35(11):3272-3277.

doi: 10.1038/s41375-021-01199-0. Epub 2021 Mar 10. Clinical characteristics and outcomes of B-ALL with ZNF384 rearrangements: a retrospective analysis by the Ponte di Legno Childhood ALL Working Group

Shinsuke Hirabayashi  1 , Ellie R Butler  2 , Kentaro Ohki  3 , Nobutaka Kiyokawa  3 , Anke K Bergmann  4 , Anja Möricke  5 , Judith M Boer  6   7 , Hélène Cavé  8 , Giovanni Cazzaniga  9 , Allen Eng Juh Yeoh  10 , Masashi Sanada  11 , Toshihiko Imamura  12 , Hiroto Inaba  13 , Charles Mullighan  13 , Mignon L Loh  14 , Ulrika Norén-Nyström  15 , Agata Pastorczak  16 , Lee-Yung Shih  17 , Marketa Zaliova  18 , Ching-Hon Pui  13 , Oskar A Haas  19 , Christine J Harrison  2 , Anthony V Moorman  2 , Atsushi Manabe  20

Response.

Both references were added

Comment 13. 

Ref 7 is no more “Online ahead of print” but already published:  Immunophenotypic changes of leukemic blasts in children with relapsed/refractory B-cell precursor acute lymphoblastic leukemia, who have been treated with Blinatumomab. Mikhailova E, Gluhanyuk E, Illarionova O, Zerkalenkova E, Kashpor S, Miakova N, Diakonova Y, Olshanskaya Y, Shelikhova L, Novichkova G, Maschan M, Maschan A, Popov A. Haematologica. 2021 Jul 1;106(7):2009-2012. doi: 10.3324/haematol.2019.241596.

Response.

Corrected

Reviewer 2 Report

  1. Your 182 total cases did all received same number of cycles of  Blinatumomab therapy. Did all transformed cases got similar cycle of treatment? 
  2. The finding of the class switch in majority of KMT2A rearranged ALL is a great observation.  However, one case with TCF3-ZNF384 cannot be predictor  of the class switch.  
  3. The TCF3-ZNF384 rearrangement have been reported in both Mixed phenotype and adult B-ALL. In the initial diagnostic flow cytometry study was any indication of mixed phenotype?
  4.  Since for whole exome studies specimen with higher percentage of blasts are required to prevent false negative results, it's useful to show a table with percentage of blasts for pre/post samples and specifically for case #1 and #3  with no mutations.
  5. The only genomic pattern observed was between Pt# 4 and 5 with KRAS, which is a common finding seen in many of the B-ALL cases. What was the prevalence of KRAS mutations in non-class switched cases post therapy? What was the prevalence of TP53 in those cases?
  6.  Finding aberration switch after treatment how would effect prognosis, and further treatment decisions?

Author Response

Referee #2 (Comments to the Author):

Response to Referee 2

Comment 1.

Your 182 total cases did all received same number of cycles of  Blinatumomab therapy. Did all transformed cases got similar cycle of treatment?

Response.

Among described switched cases one patient (#1) received two complete blinatumomab cycles. In others one complete cycle (n=2) was given and in three cases the first blinatumomab course was interrupted with lineage switch. This is shown in the text and in Table 1. The idea of the study was not to show the structure of relapses in any protocol including blinatumomab, but to show incidence and molecular patterns of lineage switch that can occur in case of blinatumomab administration in all possible situations. From our data one can see that lineage conversion could develope in different ways, as resistance or relapse, not only in KMT2A-r patients, with additional mutations and without them etc. We have not focused on the remaining patients and their treatment results because this is the area of another study. Moreover, they were treated in different ways and it is not easy to put such information to the article containing so detailed description of six switched cases. We wanted just to focus attention on the possible types of lineage switch and molecular features of the lineage conversion.

Comment 2.

The finding of the class switch in majority of KMT2A rearranged ALL is a great observation.  However, one case with TCF3-ZNF384 cannot be predictor of the class switch. 

Response.

We do realize that a single case is not a predictor, however, lineage switches in ALL with ZNF384 rearrangement were described in literature – see Oberley et al., 2018. Besides, ZNF384 rearrangement comprises a substantial fraction of B+myelo mixed phenotype leukemias (Alexander et al., 2018, doi: 10.1038/s41586-018-0436-0) and is observed in BCP-ALL with monocytic switch during the treatment (Novakova et al., 2021, 10.3324/haematol.2020.250423). Thus, we speculate that lineage switch is to be anticipated in this genetic subgroup regardless of more or less prominent signs of mixed phenotype at diagnosis”. We have added few references and modified our statement about ZNF384 fusions (lines 257-259).

Comment 3.

The TCF3-ZNF384 rearrangement have been reported in both Mixed phenotype and adult B-ALL. In the initial diagnostic flow cytometry study was any indication of mixed phenotype?

Response.

We have only seen the small additional myeloid population (near 3% of cells) which was detectable during the whole history of relapse treatment. This is indicated in both the text and tables. We cannot consider such situation as the typical bilineage MPAL because of very low proportion of myeloid blasts, although we also cannot ignore these cells.

Comment 4.

Since for whole exome studies specimen with higher percentage of blasts are required to prevent false negative results, it's useful to show a table with percentage of blasts for pre/post samples and specifically for case #1 and #3  with no mutations.

Response.

We have added table S5 to show the tumor cells percentage in described samples. Anyway, we reported all variants with VAF above 1%. Thus, we consider our data representative.

Comment 5.

The only genomic pattern observed was between Pt# 4 and 5 with KRAS, which is a common finding seen in many of the B-ALL cases. What was the prevalence of KRAS mutations in non-class switched cases post therapy? What was the prevalence of TP53 in those cases?

Response.

Unfortunately, we did not perform mutational screening for primary BCP-ALL routinely. We only have very limited cohorts collected to date: BCP-ALL with various hypodiploidy analyzed by Sanger and MLPA for TP53, where TP53 demonstrated loss-of-function variants in 2 of 8 cases (25%) and B-other ALL analyzed by RNAseq, where KRAS pathogenic variants were found in 2 of 13 cases (15.4%) and TP53 loss-of-function variants were found in 1 of 13 cases (7.7%). Neither of these groups represents a correct comparison cohort for lineage switch patients so was not mentioned in the paper. We can only stick to published data. We have added such statement to the text (lines 289-291)

Comment 6.

Finding aberration switch after treatment how would effect prognosis, and further treatment decisions?

Response.

Unfortunately, it is still difficult to define prognostic consequences of lineage switch. Of course, if switch occurs as the type of resistance to blinatumomab, the outcome is very poor despite of all available treatment options. Contrary, in patients who achieved remission after immunotherapy, lineage switch could be cured (in our series in one patient out of two) with myeloid-style treatment (and HSCT). Moreover, application of such treatment after blinatumomab can prevent relapse even in cases with pre-existed myeloid tumor population (J. Bartram et al, BJH, DOI: 10.1111/bjh.17707). We have added this statement to the end of Discussion section (lines 323-328).